# Discovering Plum, Watermelon and Grape Cultivars Founded in a Middle Age Site of Sassari (Sardinia, Italy) through a Computer Image Analysis Approach

**DOI:** 10.3390/plants11081089

**Published:** 2022-04-16

**Authors:** Marco Sarigu, Diego Sabato, Mariano Ucchesu, Maria Cecilia Loi, Giovanna Bosi, Oscar Grillo, Salvador Barros Torres, Gianluigi Bacchetta

**Affiliations:** 1Centro Conservazione Biodiversità (CCB), Dipartimento di Scienze della Vita e dell’Ambiente (DISVA), Università degli Studi di Cagliari, Viale Sant’Ignazio da Laconi, 13, 09123 Cagliari, Italy; msarigu@unica.it (M.S.); loimc@unica.it (M.C.L.); oscar.grillo.mail@gmail.com (O.G.); bacchet@unica.it (G.B.); 2Departament de Prehistòria i Arqueologia, Facultat de Geografía i Historia, Universitat de València, Av. Blasco Ibáñez, 28, 46010 Valencia, Spain; diegosabato@libero.it; 3Institut des Sciences de l’Évolution (ISEM), Centre National de la Recherche Scientifique (CNRS) Université Montpellier, UMR 5554, CEDEX 05, 34095 Montpellier, France; 4Laboratorio di Palinologia e Paleobotanica, Dipartimento di Scienze della Vita, Università di Modena e Reggio Emilia, Via Campi, 287, 41125 Modena, Italy; giovanna.bosi@unimore.it; 5Laboratório de Análise de Sementes, Departamento de Ciências Agronômicas e Florestais, Universidade Federal Rural do Semi-Árido (UFERSA), Av. Francisco Mota, Bairro Costa e Silva, 572, Mossoró 59625-900 , Brazil; sbtorres@ufersa.edu.br

**Keywords:** seed image analysis, Middle Ages context, *Citrullus*, *Prunus*, *Vitis*, Sardinia

## Abstract

The discovery of several waterlogged plant remains in a Middle Ages context (1330–1360 AD) in Sassari (NS, Sardinia, Italy) enabled the characterisation of archaeological plum fruit stones and watermelon and grape seeds through computer image analysis. Digital seed/endocarp images were acquired by a flatbed scanner and processed and analysed by applying computerised image analysis techniques. The morphometric data were statistically elaborated using stepwise linear discriminant analysis (LDA), allowing comparisons among archaeological remains, wild populations and autochthonous cultivars. Archaeological samples of plum were compared with 21 autochthonous cultivars of *Prunus domestica* from Sardinia, while archaeological watermelon seeds were compared with 36 seed lots of *Citrullus* from Europe, Africa and Asia. Moreover, archaeological grape seeds were compared with 51 autochthonous traditional cultivars of *Vitis vinifera* subsp. *vinifera* from Sardinia, 16 cultivars from Tuscany, six cultivars from Liguria, and eight cultivars from Catalonia (Spain). Archaeological plum remains showed morphological affinity with five cultivars of Sardinia. Seed features of the archaeological watermelon remains demonstrated affiliation with a proper sweet dessert watermelon, *Citrullus lanatus*, and similarity with some Sardinian cultivars. Regarding the archaeological remains of grape, morphometric comparisons showed a high similarity with autochthonous cultivars from Catalonia and Liguria. This study provides new information about ancient fruit cultivated and consumed during the Middle Ages in Sardinia.

## 1. Introduction

In recent years, archaeobotany in Italy has developed remarkably and has been documented by several scientific studies and projects [1,2].

Currently, archaeobotanical research in Sardinia (Italy) is focused on the Neolithic [3], Chalcolithic [4], Bronze Age [5,6,7,8,9,10,11], and Phoenician and Punic periods [12,13,14,15]. Unfortunately, only the archaeobotanical studies of Becca et al. [16] and Bosi and Bandini Mazzanti [17] refer to the Middle Ages in Sardinia, in contrast to many Middle Age sites investigated in the Italian peninsula, which represent good examples of seed/fruit information for this period [17,18,19,20,21,22,23,24,25,26,27,28,29].

Generally, plant macroremains are identified by comparing archaeobotanical materials with modern reference collections or by using images of seeds/fruit reported in several atlases [30,31,32,33,34,35].

Over the last two decades, to overcome the manual discrimination system of seeds/fruit, image analysis has received considerable attention in plant research using automated systems that have the potential to replace human visual assessments. This methodology was successfully applied to discriminate wild and cultivated plants [36,37,38,39,40,41,42,43,44,45,46] and to identify archaeobotanical remains [6,7,9,15,45,46,47,48,49,50,51].

The exceptional and well-preserved quantity of seeds/fruit stones of plum, watermelon, and grape found in a well of Via Satta in the historical city centre of Sassari (NW Sardinia) represents an excellent opportunity to investigate ancient cultivars present in the Middle Ages in Sardinia. 

In the light of the above, the main goals of this study are to compare archaeological seeds/fruit stones with modern cultivars using image analysis techniques and to gain new information about the cultivation and consumption of fruit during the Middle Ages, in Sardinia.

## 2. Archaeological Background

The development of the city of Sassari, today the second largest city in Sardinia, took place during the 12th and 13th centuries AD, when it became one of the main centres for inland and foreign trade of a wide variety of products. In 1272, the city fell under the control of the Republic of Pisa and later, in 1294, under the Republic of Genoa [52]. In 1323, citizens decided to become part of the Crown of Aragon [53]. This alliance did not last long, since a few years later, between 1330 and 1331, the continuous insurrections of the city led the Aragonese to the expulsion of some local citizens, which were soon replaced by colonists from Catalonia, Aragona, Valencia, and Tarragona [54]. The events that occurred in this historical phase are highly important for the development of the city of Sassari in the Middle Ages.

During the renovation of the Via Satta, in 2007, in the core of the city centre of Sassari, a Middle Ages well was discovered (Figure 1a,b). The Middle Ages structure was originally part of an open area or domestic courtyard, which has been dated back to the period between 1330 and 1360 AD, according to the typology of majolica fragments from Pisa, Savona, and Valencia widely diffused in that period [55]. The well had a diameter of only 90 cm and a depth of 14 m (Figure 1c). The anoxic conditions and the constant presence of water allowed the optimal conservation of the biological remains discovered. The sediment inside the well was rich in waterlogged plant remains, including wood, seeds, and fruits. A total of 880,000 items (about 117 taxa) have been identified previously by Bosi and Bandini Mazzanti [17] and Bertacci [56]. A significant well-preserved amount of plum, watermelon, and grape remains was recovered, and for this reason, it was decided to investigate these important taxa.

## 3. Results

### 3.1. Plums Fruit-Stones

To identify the phenotypic characteristics of the archaeological remains identified as *Prunus domestica*, in the previous study by Bosi and Bandini Mazzanti [17] and Bertacci [56], the 71 archaeological fruit-stones were compared with the 21 modern cultivars present in Sardinia. Based on this comparative analysis, the LDA showed an overall percentage of correct identification of 64.9% (Table 1); the result showed that the archaeological fruit-stones were matched to five modern cultivars, in particular with the cultivar ‘Croccorighedda’ (CRO) with a classification percentage of 35.2% (Table 1). The other fruit-stones were assigned to the cultivars ‘Gialla di Bosa’ (GIB; 16.9%), ‘Cariasina’ (CAR; 12.7%), ‘Sanguigna di Bosa’ (SBO; 9.9%), and ‘Laconi F’ (LA5; 8.5%) (Table 1).

### 3.2. Watermelon Seeds

In the first analysis, the archaeological watermelon seeds were compared to 1039 seeds of *Citrullus colocynthis*, 1712 seeds of *C. lanatus* var. *lanatus*, and 487 seeds of *C. lanatus* var. *citroides*. The overall percentage of correct identification reached 93.8%, and the archaeological seeds, included in the analysis as unknown items, were identified as *C. lanatus* var. *lanatus* in 95.7% of the cases (Table 2). 

In the first analysis, the archaeological watermelon seeds were compared to 1039 seeds of *C. colocynthis*, 1712 seeds of *C. lanatus* var. *lanatus*, and 487 seeds of *Citrullus lanatus* var. *citroides*. The overall percentage of correct identification reached 93.8%, and the archaeological seeds, included in the analysis as unknown items, were identified as *C. lanatus* var. *lanatus* in 95.7% of the cases (Table 2). 

The archaeological samples identified as *C. lanatus* var. *lanatus* were compared with the group of modern specimens of *C. lanatus* var. *lanatus*. The discriminant analysis showed a high percentage of allocation on the group of samples from Italy. In particular, the archaeological samples matched with the traditional cultivars from Sardinia, in particular with the cultivar known as ‘Sindria Gialla’ (LnITS3), with an allocation percentage of 22.9% (Table 3). The remaining samples found a morphological affinity with ‘Sindria di Gonnos’ (LnITS4), ‘Sindria di Carloforte’ (LnITS2), and ‘Sindria Bianca’ (LnITS1) (Table 3, Figure 2).

The other archaeological samples showed affinity with cultivars from southern Spain (LnES65), Uzbekistan (LnUZ78), Kyrgyzstan (LnKZ81), Algeria (LaDZ59), and Angola (LnAO53) (Table 3, Figure 2).

### 3.3. Grape Seeds

Archaeological *Vitis vinifera* seeds were added to the classifier as an unknown group and compared with 81 modern grape cultivars from Tuscany, Liguria, Catalonia, and Sardinia. In the LDA first analysis, the archaeological materials showed a strong morphological similarity with the Catalonia cultivar groups with a percentage of 56.4% and with the cultivars from Liguria with a percentage of 26.3% (Table 4, Figure 2). The other archaeological grape seeds were assigned to Sardinian (10.2%) and to the Tuscany cultivars (7.0%) (Table 4, Figure 2).

In order to understand which modern grape cultivar showed a close relationship with the archaeological seeds, another analysis was conducted. The archaeological grape seeds attributed to the Catalonia cultivars were compared with the modern cultivars currently present in this region. The results showed a high similarity to ‘Garnacha’, with a percentage of 28%, while the remaining grape seeds were assigned to ‘Sumoll’ (25.5%), ‘Monastrell’ (22.2%), ‘Macabeo’ (15.3%), and ‘Ull de Llebre’ (8.0%) (Table 5, Figure 2). Similarly, the archaeological grape seeds that were assigned to the group of grapes from Liguria were attributed to cultivars ‘Lumassina’ (50.3%), ‘Rossese’ (36.3%), and ‘Bianchetta Genovese’ (8.7%) (Table 5, Figure 2). The archaeological samples assigned to Sardinian cultivars referred in particular to six cultivars never exceeding 10% of correct classification (Table 5, Figure 2). Finally, the archaeological grape seeds assigned to Tuscany were allocated a high percentage to the cultivar ‘Bracciola nera’ (37.6%) (Table 5, Figure 2).

## 4. Discussion

The image analysis system applied to archaeological seed remains discovered in the well of Via Satta confirmed the presence of different fruit cultivars belonging to *Prunus domestica*, *Citrullus lanatus*, and *Vitis vinifera* subsp. *vinifera*.

The archaeological plum remains showed morphological affinity with five ancient cultivars of *P. domestica* (Croccorighedda, Gialla di Bosa, Cariasina, Sanguigna di Bosa, and Laconi F), with yellow and violet skin colour, currently cultivated in the territories of Bosa, Laconi, and Nuoro. The results suggest that these cultivars were already cultivated in Sardinia during the Middle Ages. Most of the samples of plum remains from the Medieval well showed a close correlation with Croccorighedda, an ancient cultivar grown in Sardinia two centuries ago and mentioned in the bibliography [57,58]. The characteristics of its fruit are obovate in shape, and the colour of its skin is bright yellow. Another interesting cultivar identified in the Medieval well is Sanguigna di Bosa, a cultivar documented in the Punic period in Sardinia [15], indicating interest in the cultivation of this cultivar for over 1800 years, and it is still grown today by small custodian Sardinian farmers. 

Regarding watermelon, morphometric analysis confirmed that the archaeological remains likely belong to the proper sweet dessert watermelon instead of citron melon or wild colocynth. Most Sardinian autochthonous cultivars, together with some Spanish, African, and Central Asian accessions, showed a relationship with the archaeological seeds, indicating a common lineage. The introduction of sweet dessert watermelon in Europe is complex and unclear. Recently, Paris [59] concluded that it was selected in the Mediterranean Basin by no later than the 2nd century AD, while Watson [60] sustained the hypothesis that it was introduced during the Islamic period after being selected and improved in India. In Europe, archaeological records of watermelon have been reported for several Greek and Roman sites [22,61], and two seeds were discovered in Sardinia in Phoenician and Punic contexts [50]; however, it is difficult to determine which typology they belonged to. In fact, according to Paris [59], the citron melon arrived in Mediterranean lands during or prior to the Roman period, and Megaloudi [61] suggested that primitive bitter forms of watermelon might have existed in the pre-Islamic era. In any case, regardless of whether this crop arrived in Europe for the first time prior to the Islamic period, it is likely that it had a limited diffusion before the 13th century AD [61]. Bates and Robinson [62] also evidenced that historical sources of watermelons before the 16th century AD were sparse and that, thereafter, a wide range of watermelon cultivars were decrypted. Molecular analysis of ancient watermelon seeds also confirmed the presence of different cultivars during the Middle Ages and the Renaissance [63,64]. Some watermelon typologies were also selected for the consumption of seeds instead of fruit [65]. Linguistically, the etymology of the word watermelon in Spanish and Catalan, sandía and síndria, respectively, has an Arabic origin, literally meaning “from the region of Sind,” Pakistan, according to Real Academia Española (RAE) [66] and Gran Diccionari de la Llengua Catalana (GDLC) [67]. The Sardinian name of watermelon, sìndria (with local variations of sìndia, srìndia, cindria), only in limited areas called foràstigu and patecca [68], is also related to Spanish and Catalan names. As detailed in the introduction, at the time the city of Sassari was part of the Crown of Aragon, some of the Iberian territories were dominated by Muslims for several centuries [69]. The affinity found among the archaeological remains with Sardinian, Spanish, African, and Central Asian accessions might be linked to the different historical events that took place on the island before and during the Middle Ages.

Regarding archaeological grape seeds, morphometric analysis showed similarities with autochthonous cultivated grapes from Catalonia, in particular with Garnacha, Sumoll, and Monastrell, and from Liguria with Lumassina and Rossese cultivars. The seed image analysis established that archaeological grape seeds belonged to different cultivars, probably used for wine production in Medieval times.

The Garnacha cultivar is present in Sardinia with the name Cannonau, which is a synonym for the Spanish cultivar Garnacha [70]. Cannonau is the most cultivated grape in Sardinia (its oldest mention dates back to 1549), and the origin of this ancient vine is still under study [71]. The presence of Garnacha in the well of Via Satta is probably due to an ancient bond dating back to the period in which Sardinia was a Spanish colony (1479–1714 AD), as during this period, the two areas had intense commercial exchanges [72]. We still do not fully know whether Cannonau was already widespread in Sardinia before Spanish colonisation or whether it was brought to Sardinia by the Spanish under the name Garnacha. According to Labra and De Mattia [70], the hypothesis that the Spanish found some interesting cultivars and imported them home for consumption cannot be discarded; in any case, our analyses seem to confirm their presence in Sardinia, at least since around 1300 AD.

Sumoll is an ancient vine mentioned for the first time in 1797 AD [73] and is currently grown mainly in the Catalan region, while its synonymy Vijariego Negro is grown in the Canary Islands [73]. There is no information about its presence in Sardinia in the past, and at present, there is no vine on the market with this name or its synonymy cultivated on the island. The results also showed the presence of Monastrell, an ancient grape cultivar mentioned in 1381 AD by Francesc Eiximenis in Empordà [73]. Additionally, in this case, no historical documentation of its presence in Sardinia in the past is available. Our analyses also indicated the presence of the Macabeo cultivar among the archaeological samples. Macabeo is a cultivar originating from Vilafranca del Penedès in Catalonia, and its oldest mention dates back to the early 17th century AD by Fray Miquel Agustí [73]. We do not have any historical information about this cultivar’s introduction in Sardinia. In the archaeological samples, two other cultivars of Ligurian grapes, Lumassina and Rossese, also emerged. These two varieties, originally from Liguria, are currently not present in Sardinia, and at the moment, there is not much information on their origin. The most reliable hypothesis is that these two cultivars were imported and cultivated in Sardinia during the government of the city of Sassari by the Republic of Genoa, and over time, their cultivation was lost. 

Of all the cultivars analysed, those from Sardinia showed the lowest percentage of correlation with the Medieval grape seeds, while only one cultivar from Tuscany seemed to have a good chance of being present at the time of Pisan domination in Sardinia. It is the Bracciola nera, a grape cultivar mentioned by historical sources for the first time in 1600 AD by Soderini and in 1825 AD by Acerbi, who identifies it in Liguria with the name Braciola in Cinque Terre [73].

## 5. Materials and Methods

### 5.1. Archaeological Samples

A total of 71 plum fruit stones, 70 watermelon seeds, and 1444 grape pips from the well of Via Satta were analysed for this study by seed image analysis (Figure 3). 

The archaeological samples were previously cleaned and identified at the species level at the Laboratorio di Palinologia e Paleobotanica of the University of Modena and Reggio Emilia and then delivered to Cagliari University for subsequent morphometric analysis to determine the cultivars.

### 5.2. Modern Plant Materials

Samples of modern plum fruit stones (*Prunus domestica* L.), referring to 21 autochthonous Sardinian cultivars (Appendix A), were collected from the ISPA field catalogue located in Nuraxinieddu (Oristano, Sardinia). In this study, the same modern samples published in Ucchesu et al. [15] used for the characterisation of plum remains found in the Phoenician and Punic sites of Santa Giusta were considered. A total of 1290 modern plum fruit stones were used as comparison materials (Appendix A).

For the reference collection of the genus *Citrullus* Schrad. ex Eckl. & Zeyh., we selected three taxa: the wild colocynth (*C. colocynthis* (L.) Schrad.), the common sweet dessert watermelon (*C. lanatus* (Thunb.) Matsum. & Nakai var. *lanatus*), and the citron or tsamma melon (*C. lanatus* var. *citroides* (L.H.Bailey) Mansf.). The modern reference collection included 36 accessions from Europe, Africa, and Asia: 20 cultivars of *C. lanatus* var. *lanatus*, five cultivars of *C. lanatus* var. *citroides*, and 11 accessions of *C. colocynthis* (Appendix A). Accessions were provided by COMAV Genebank (Centro de la Conservación y Mejora de la Agrodiversidad Valenciana, Valencia, Spain), Bari GeneBank IBBR/CNR, the Agriculture Department of the University of Sassari, and the AGRIS Agency (Agenzia per la Ricerca in Agricoltura della Regione Sardegna) (Appendix A).

Regarding grape, the modern reference collection was composed of 51 autochthonous traditional cultivars of *Vitis vinifera* L. subsp. *vinifera* from Sardinia, 16 cultivars from Tuscany, six cultivars from Liguria, and eight cultivars from Catalonia (Spain) (Appendix A). Sardinia grape cultivars were obtained from the AGRIS germplasm field collection at Ussana (Sardinia) (Appendix A). Tuscany and Liguria grape cultivars were obtained from the Agricultural Research Council—Viticulture Research Centre (CRA-VIT) of Conegliano Veneto (Italy) (Appendix A). A total of 8587 modern grape seeds were used in this study.

### 5.3. Seed Image and Statistical Analysis

Digital images of archaeological and modern samples were acquired at the Sardinian Germplasm Bank (BG-SAR), using a flatbed scanner (Epson Perfection V550 photo), with a digital resolution of 400 dpi for a scanning area not exceeding 1024 × 1024 pixels [36]. Each accession was scanned twice, first with a white and then a black background. The images were segmented using the software package ImageJ v. 1.53n (http://rsb.info.nih.gov/ij) (accessed on 11 January 2019), and the plugin Particles8, freely downloadable on the official website (http://www.mecourse.com/landinig/software/software.html), (accessed on 1 January 2019) was used to measure 26 morphometric parameters (Table 6).

Regarding the statistical processing, the analysis was performed using the IBM SPSS (Statistical Package for Social Science) software version 16.0 [74], applying the Linear Discriminant Analysis (LDA).

To verify the performance of the LDA a cross-validation procedure was applied considering three statistical variables *Tolerance*, *F-to-enter*, and *F-to-remove* following the procedure described in detail in Sarigu et al. [44].

## 6. Conclusions

The morphometric study carried out in this work constitutes an innovative contribution to characterising past agrobiodiversity in addition to providing new information about the phenotypic characteristics of fruit cultivated in Sardinia in the Middle Ages.

Morphometric characterisation indicated that the cultivation and use of *Prunus domestica* were well established in Sardinia during the Middle Ages and that watermelon seeds belong to a proper sweet dessert *Citrullus lanatus* var *lanatus*, closely linked to modern Sardinian cultivars. The close connection of these archaeological seeds and the Sardinian cultivars with Spanish, African, and Central Asia typologies was also found, and historical events may also have played an important role in their similarity. 

Additionally, new knowledge about the grape history of Sardinia was acquired, enabling us to obtain information about the trade between Sardinia, Spain, and mainland Italy. The history of viticulture in Sardinia is complex, and only morphological and molecular analyses of the various cultivars, together with historical sources, can trace the habits of the people. Sardinia has a long history of grape domestication and may have been a secondary grape domestication centre since the Bronze Age [8]. For these reasons, the results obtained through seed image analysis represent an important step of knowledge on the ancient plum, watermelon and grape cultivars cultivated in Sardinia. 

## Figures and Tables

**Figure 1 plants-11-01089-f001:**
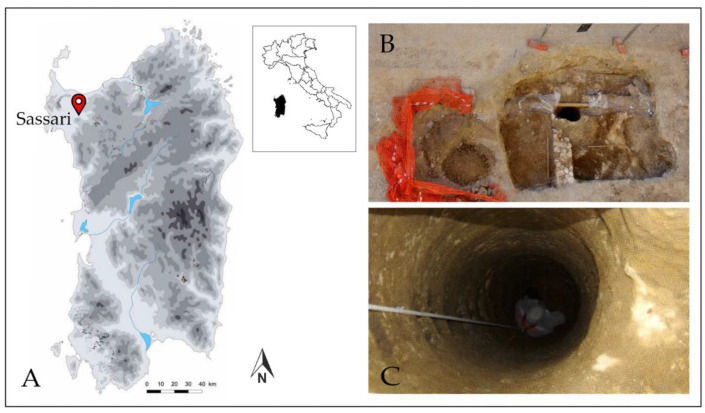
(**A**) Location of the city of Sassari; (**B**) The archaeological excavation area of Via Satta (Sassari, NW Sardinia); (**C**) The Middle Ages well (Biccone 2013, modified).

**Figure 2 plants-11-01089-f002:**
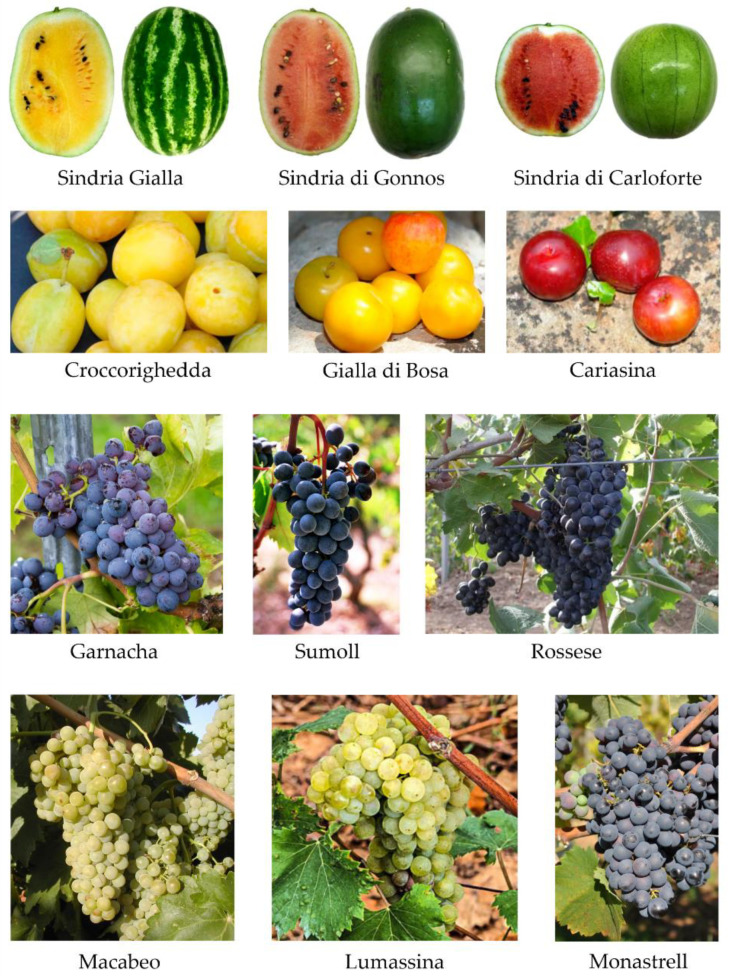
Selected modern cultivars with a close relationship with the archaeobotanical remains found in the Medieval well of Sassari.

**Figure 3 plants-11-01089-f003:**
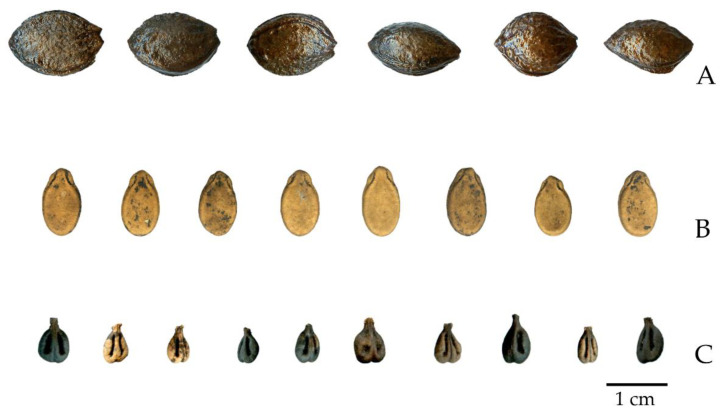
Representative image of some of the archaeological samples used in this work: (**A**) *Prunus domestica*; (**B**) *Citrullus lanatus* var. *lanatus*; (**C**) *Vitis vinifera* subsp. *vinifera*.

**Table 1 plants-11-01089-t001:** LDA percentage results of comparison among modern *Prunus domestica* cultivars and Middle Ages fruit-stones considered as unknown.

Code	Cultivar Name	N° Seeds	Cultivar Classification (%)	Archaeological Seeds Classification (%)
CAD	Cariadogia	96	69.8	-
CAR	Cariasina	39	64.1	12.7
COL	Columbu	60	58.3	1.4
COR	Coru	79	54.4	1.4
CRO	Croccorighedda	98	75.5	35.2
BON	Bonarcado	99	67.7	1.4
DOR	Dore	23	47.8	1.4
GIB	Gialla di Bosa	99	89.9	16.9
LA2	Laconi B.	99	59.6	5.6
LA4	Laconi E.	30	83.3	-
LA5	Laconi F.	30	73.3	8.5
LIM	Limuninca	30	63.3	-
MEL	Meloni	60	83.3	-
LAR	Laconi Rosata	25	80.0	1.4
NES	Nera Sarda	69	29.0	1.4
LA3	Ollanu de Ou	72	79.2	-
PAR	Paradisu	15	86.7	-
SAE	Sant’Elia	29	79.3	-
SBO	Sanguigna di Bosa	100	69.3	9.9
SIG	Sighera	99	72.7	1.4
SAG	San Giovanni	39	46.2	1.4
Overall				64.9%

**Table 2 plants-11-01089-t002:** LDA percentage results of comparison among modern seeds of *Citrullus colocynthis*, *C. lanatus* var. *lanatus*, and *C. lanatus* var. *citroides* and Middle Ages watermelon seeds considered as unknown.

	*C. colocynthis*	*C. lanatus* var. *lanatus*	*C. lanatus* var. *citroides*	Total
*Citrullus colocynthis*	89.1	8.0	2.0	100.0
*C. lanatus* var. *lanatus*	0.7	97	2.0	100.0
*C. lanatus* var. *citroides*	1.8	6.4	91.8	100.0
Archaeological samples	-	95.7	4.3	100.0
Overall				93.8%

**Table 3 plants-11-01089-t003:** Correct classification percentage between *Citrullus lanatus* var. *lanatus* cultivars and Middle Ages seeds considered as unknown.

CODE	Country	Locality	Cultivar Name	Cultivar Classification (%)	Archaeological Classification (%)
LnAO22	Angola	Namibe	Unknown	73.1	-
LnAO53	Angola	Luanda	Unknown	70.7	11.4
LnAO55	Angola	Luanda	Unknown	80.9	1.4
LnDZ25	Algeria	Mostaganem	Unknown	43.5	-
LnDZ59	Algeria	Ammes, Béchar	Unknown	64.9	-
LnDZ78	Algeria	Mostefa Ben Brahim	Unknown	55.1	-
LnES51	Spain	Rota, Cádiz	Sandía de Rota	86.5	-
LnES62	Spain	Moraleda, Granada	Sandía inverniza	73.1	-
LnES65	Spain	Huelva	Sandía de verano	52.7	7.1
LnES81	Spain	Mallorca	Sandía de pinyol blanc	78.5	-
LnGR32	Greece	Navplion, Argolide	Karpusi	72.2	-
LnITS1	Italy	Benetutti, Sardinia	Sindria bianca	60.0	4.3
LnITS2	Italy	Carloforte, Sardinia	Sindria	68.0	11.4
LnITS3	Italy	Sant’Antioco, Sardinia	Sindria gialla	57.0	22.9
LnITS4	Italy	Gonnos, Sardinia	Sindria	34.0	15.7
LnITS5	Italy	Benetutti, Sardinia	Sindria niedda	59.0	-
LnKZ81	Kyrgyzstan	Dzho, Lenin	Unknown	71.9	20.0
LnMA3	Morocco	Unknown	Unknown	59.3	-
LnSY93	Syria	Damasco	Unknown	73.3	-
LnUZ78	Uzbekistan	Salar	Unknown	48.8	5.7

**Table 4 plants-11-01089-t004:** LDA percentage results of comparison among the Middle Ages grape seeds, considered as unknown and four different modern cultivars groups considering their origin.

	Tuscany	Liguria	Catalonia	Sardinia	Total
Tuscany	36.8	27.1	15.5	20.5	100.0
Liguria	23.6	35.6	10.7	30.1	100.0
Catalonia	11.8	11.9	48.2	28.2	100.0
Sardinia	10.0	12.3	16.7	61.0	100.0
Archaeological grape seeds	7.0	26.3	56.4	10.2	100.0
Overall					52.3%

**Table 5 plants-11-01089-t005:** Correct classification percentage between *Vitis vinifera* subsp. *vinifera* cultivars and Middle Ages seeds considered as unknown.

Region	Cultivar Name	Cultivar Classification (%)	Middle Ages Seeds Classification (%)	N° of Archaeological Seeds Allocation
Tuscany	Bracciola Nera	37.6	37.6	76
	Canaiolo Bianco	41.1	12.9	
	Canaiolo Nero	60.2	13.9	
	Livornese Bianca	66.1	10.9	
Liguria	Bianchetta Genovese	62.6	8.7	362
	Lumassina	85.3	50.3	
	Rossese	62.7	36.3	
Catalonia	Macabeo	72.4	15.3	803
	Sumoll	65.4	25.5	
	Garnacha	66.3	28.0	
	Monastrell	69.7	22.2	
	Ull de Llebre	57.6	8.0	
Sardinia	Galoppu	47.5	5.4	65
	Malvasia di Sardegna	41.0	6.1	
	Nuragus	41.3	8.8	
	Bianca addosa	30.3	6.1	
	Licronaxu rosa	44.0	10.8	
	Gabriella	34.0	6.8	

**Table 6 plants-11-01089-t006:** List of morphometric features measured on each seed/fruit-stone.

Features	Description
Perim	Perimeter, calculated from the centres of the boundary pixels
Area	Area inside the polygon defined by the perimeter
Pixels	Number of pixels forming the endocarp image
MinR	Radius of the inscribed circle centred at the middle of the seed
MaxR	Radius of the enclosing circle centred at the middle of the seed
Feret	Largest axis length
Breadth	Largest axis perpendicular to the Feret
CHull	Convex hull or convex polygon calculated from pixel centres
CArea	Area of the convex hull polygon
MBCRadius	Radius of the minimal bounding circle
AspRatio	Aspect ratio = Feret/Breadth
Circ	Circularity = 4·π·Area/Perimeter^2^
Roundness	Roundness = 4·Area/(π·Feret^2^)
ArEquivD	Area equivalent diameter = √ ((4/π)·Area)
PerEquivD	Perimeter equivalent diameter = Area/π
EquivEllAr	Equivalent ellipse area = (π·Feret·Breadth)/4
Compactness	Compactness = √ ((4/π)·Area)/Feret
Solidity	Solidity = Area/Convex_Area
Concavity	Concavity = Convex_Area-Area
Convexity	Convexity = Convex_hull/Perimeter
Shape	Shape = Perimeter^2^/Area
RFactor	RFactor = Convex_Hull /(Feret·π)
ModRatio	Modification ratio = (2·MinR)/Feret
Sphericity	Sphericity = MinR/MaxR
ArBBox	Area of the bounding box along the feret diameter = Feret·Breadth
Rectang	Rectangularity = Area/ArBBox

## Data Availability

All the data supporting this study are included in the article.

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
