# Peer review of "Discovering Plum, Watermelon and Grape Cultivars Founded in a Middle Age Site of Sassari (Sardinia, Italy) through a Computer Image Analysis Approach"

_plants, 2022, doi:10.3390/plants11081089_

Round 1
Reviewer 1 Report
The authors of this manuscript present an interesting research study on discovering plum, watermelon and grape cultivars founded in a Middle Age site of Sassari (Sardinia, Italy) through a computer image analysis approach. The presented tables and figures are clear. The authors conclude the findings of their work in a discussion section. The text needs few revisions. However, this research is also archeological research I believe that it can add further research since it focuses in plants.
Abstract
COMMENT:
According to my opinion the Abstract describe sufficiently the findings of this work. Abstract should be in one paragraph.
Introduction
Introduction section is well written and, in my opinion, give the appropriate information without being extended.
Line 70-87 I am not sure if this information is needed
Please check justification of the text on page 2.
Results
Line 99 P. domestica
Line 133 Citrullus lanatus
Discussion
Line 182 Paragraph
Conclusions
Check paragraph style
References
COMMENT:
Please check carefully the reference list according to the author’s instruction once again in order to be sure that is correctly typed and mentioned within the text.
Author Response
Dear Editor and Reviewer,
On behalf of all authors, I am grateful for your feedback, very helpful to improve the quality of the manuscript.
We carefully modified the text according to your proposed revisions, please find enclosed our manuscript.
Thanks in advance for your attention, we look forward to hearing from you,
Sincerely
Mariano Ucchesu
Reviewer 2 Report
Comments on the manuscript titled „Discovering plum, watermelon and grape cultivars founded in a Middle Age site of Sassari (Sardinia, Italy) through a computer image analysis approach” by Marco Sarigu, Diego Sabato, Mariano Ucchesu, Maria Cecilia Loi, Giovanna Bosi, Oscar Grillo, Salvador Barros Torres, Gianluigi Bacchetta submitted to section: Plant Systematics, Taxonomy, Nomenclature and Classification.
The article describes very interesting research on the origin and history of watterlogged plants in Sardinia and fits perfectly into the Special Issue: Crops and Agriculture in Medieval Age in Europe. The authors look for taxonomic/genetic links between varieties used in the Middle Ages and modern varietes of plum, watermelon and grape cultivars grown in Italy. However, the title of the article suggests a broad application of image analysis. In this regard, I feel somewhat unsatisfied because the technique of image collection and processing is not described well enough for me to be able to repeat these experiments. he details of the comparative analysis are very interesting. How for example a rating of 64.9% was obtained? Such details are in fact the main part of the description of the computer analysis approach. It is not clear how the images were obtained as in Fig.3. Were the images taken on a white background? How was the background filtered? Is figure 3 simply a composite of cut-out images? Such information should be included in the manuscript. It would be good additionally in supplementary material to show the unprocessed scans in relation to the digitally processed ones.
Table 6 lists of the measured morphometric traits. It would be good to describe in detail how these features were collected (provide an algorithmor describe how the Canning system works?) Perimeter2 and not Perimeter2; Feret2 and not Feret2 etc.
Tables 1 and 2 does Archeological seeds classification mean „Similarity”? Than what the „Cultivar classification (%)” means? Please add the description to Materials and methods section.
The numbering of literature references is not continuous. On line 60 there is item [51] followed by item [55]. Items [52-54] appear later in the text.
Shift the Information from lines 383-390 „The work of M. Ucchesu has received funding from the European Union’s Horizon 2020 research and innovation programme under the Marie SkÅ‚odowska-Curie grant agreement (No 101019563 – VITALY).” to „Funding:” placed in Line 373.
Two of the authors have the same letter abbreviations G.B (Giovanna Bossi and Gianluigi Bacchetta). Please think of 3-letter abbreviations, e.g. B.BO and G.BA. This will simplify the descriptions in the „Author Contributions:” section.
Please note the formatting of the literature items. Items 2,12,13,14,16,18,19,21,23 24, 37,48, 52, 58, 63 and 73 („Editore” possibly more cases) use periods, commas, semicolons and italics incorrectly.
S1 and S2 tables „ Localization” not „Locality”
Language/wording of the article needs minor corrections. Below are only examples from first few lines
Line 23 stones, (remove and)
Line 34 last word „Seeds” remove as it repeats in Line 35
Line 45 has developer remarkably and has been documented……..
Line 65 „In light of these elements” should be replaced with „In the light of the above”
Line 99, 183 etc. P. domestica
Author Response

(The authors gave the same response as above.)
